# Effect of Selection for Pyrethroid Resistance on Abiotic Stress Tolerance in *Aedes aegypti* from Merida, Yucatan, Mexico

**DOI:** 10.3390/insects12020124

**Published:** 2021-01-31

**Authors:** Keenan Amer, Karla Saavedra-Rodriguez, William C. Black, Emilie M. Gray

**Affiliations:** 1Department of Organismal Biology & Ecology, Colorado College, Colorado Springs, CO 80903, USA; k_amer@coloradocollege.edu; 2Department of Microbiology, Immunology & Pathology, College of Veterinary Medicine and Biomedical Sciences, Colorado State University, Fort Collins, CO 80523, USA; karla.saavedra_rodriguez@colostate.edu (K.S.-R.); william.black@colostate.edu (W.C.B.IV)

**Keywords:** climate, fitness, heat stress, insecticide resistance, mosquito, osmotic stress

## Abstract

**Simple Summary:**

*Aedes aegypti* is the principal vector of major human pathogens, including dengue, Zika, chikungunya, and yellow fever viruses. Vector control relies mostly on the use of pyrethroid insecticides that kill mosquitoes by disabling the nervous system through binding to the voltage-gated sodium channel (vgsc). Resistance mechanisms have evolved most commonly as mutations in the *vgsc* gene or in genes associated with detoxification. These mutations are thought to associate with fitness costs, such that the frequency of resistant genotypes should decrease in the absence of insecticide use, and this assumption is critical to managing resistance through insecticide rotation strategies. While most studies to date have investigated life history parameters such as fecundity, we sought to investigate whether environmental stress resistance traits might also vary with insecticide resistance. We found, contrary to our expectations, that a strain selected for enhanced insecticide resistance had higher thermotolerance than its sister insecticide susceptible counterpart. Overall, our results indicate that abiotic resistance traits can correlate with insecticide resistance in surprising and variable ways, potentially complicating the management of insecticide resistance in the field.

**Abstract:**

The study of fitness costs of insecticide resistance mutations in *Aedes aegypti* has generally been focused on life history parameters such as fecundity, mortality, and energy reserves. In this study we sought to investigate whether trade-offs might also exist between insecticide resistance and other abiotic stress resistance parameters. We evaluated the effects of the selection for permethrin resistance specifically on larval salinity and thermal tolerance. A population of *A. aegypti* originally from Southern Mexico was split into two strains, one selected for permethrin resistance and the other not. Larvae were reared at different salinities, and the fourth instar larvae were subjected to acute thermal stress; then, survival to both stresses was compared between strains. Contrary to our predictions, we found that insecticide resistance correlated with significantly enhanced larval thermotolerance. We found no clear difference in salinity tolerance between strains. This result suggests that insecticide resistance does not necessarily carry trade-offs in all traits affecting fitness and that successful insecticide resistance management strategies must account for genetic associations between insecticide resistance and abiotic stress resistance, as well as traditional life history parameters.

## 1. Introduction

The evolution of insecticide resistance is a well-known phenomenon that has been observed in many species in response to all major classes of insecticides [1,2]. Resistance can evolve via multiple mechanisms, including improved metabolic detoxification, point mutations affecting the sensitivity of target proteins, and reduced penetration of the cuticle [3,4,5,6]. Such resistance mechanisms have a significant adaptive value for the organism under conditions of insecticide use; however, they induce physiological changes that can negatively impact fitness and lead to a reversion to the susceptible phenotype in the absence of insecticide selective pressure [7,8,9]. Adding complexity to this picture, the correlated fitness effects of resistance may change over time in a population as new mutations arise and interact with others [10]. Since insecticide resistance management (IRM) strategies such as insecticide rotation rely on resistance mutations having negative fitness costs, it is critical to gain a better understanding of how these mutations correlate with life history and environmental stress resistance traits in mosquitoes. Identifying the occurrence and mechanistic origins of such fitness correlates is not only important for modeling the effects of insecticide applications on population dynamics but might also provide new tools for vector control.

The reduced fitness of resistant genotypes has been documented for numerous species and insecticides [8]. Reduced fitness was found to take the form of reduced fecundity or reproductive rate [11,12,13], higher mortality [14], lower body mass [9], and even lower rates of courtship [15]. Such fitness effects of resistance may come as a result of direct or pleiotropic effects of the resistant genes themselves, of alleles at nearby loci co-segregating with the resistance allele, or of trade-offs in resource allocation [9,16,17]. Despite the assumption that resistance often comes with a fitness trade-off, research has shown that this is not always the case; in fact, some studies have found the opposite. For example, malathion-resistant *Tribolium castaneum* (Coleoptera: Tenebrionidae) showed enhanced male reproductive success and increased female fecundity [18,19]. Additionally, a lab population of *Anopheles funestus* (Diptera: Culicidae) selected for pyrethroid resistance was found to have a higher ratio of females, a higher proportion of females that successfully produced eggs, and a higher proportion of eggs surviving to adulthood when compared to its susceptible counterpart [20]. This paints a mixed picture across species and insecticides, leading researchers to question which mechanisms promote the maintenance of resistance alleles in populations [21,22].

*Aedes aegypti* (Diptera: Culicidae) is a container dwelling mosquito species that is closely associated with urban areas throughout the tropics and subtropics and is also the principal vector of dengue, Zika, chikungunya, Rift Valley fever, and yellow fever viruses. Heavy insecticide use has led to a rapid rise in resistance genotypes worldwide—in particular, mutations in the voltage-gated sodium channel transmembrane gene (vgsc, aaeNav, LOC5567355) that are called kdr mutations (knockdown resistance) because they reduce pyrethroid binding [23,24]. Recent surveys from Southern Mexico have found that pyrethroid use has resulted in locally adapted and genetically differentiated populations [25], and that geographically distinct populations respond differently to the interruption of insecticide use with some losing resistance faster than others [26]. This provides evidence of a mosaic of dynamic resistance genotypes across the landscape, each with their unique, environment-dependent fitness costs. As in other species, studies on the fitness costs of insecticide resistance in *A. aegypti* have mostly focused on traditional life history parameters. Resistant populations have generally shown negative fitness effects, such as lower fecundity and hatchability [27,28,29], higher adult mortality rates [30,31,32], and even lower energy reserves [30]. No studies to date have examined the tolerance of *A. aegypti* to abiotic stresses found in the field. Yet, such factors, which vary in space and time, are likely to have major effects on the survival rates of alternate genotypes.

In the field, larvae and adults of *A. aegypti* are subjected to many abiotic stressors, such as thermal extremes and dehydration. Climate is the principal factor affecting the global distribution of *A. aegypti.* Studies have shown that complete larval development cannot occur at temperatures lower than 10–16 °C and above 36 °C [33,34]. Immature stages are particularly sensitive to temperature due to the fact that they live in small, isolated pools and cannot easily escape unfavorable conditions [35]. Larvae may also be subject to shifting water chemistry as the water in their habitat evaporates [36]. Adults can be subject to dehydration stress, a factor known to impact the geographical distribution, reproductive capacity, and longevity of arthropods [37,38]. In sum, the physical environment exerts a strong selective pressure on mosquitoes. Therefore, the dynamics of insecticide resistance in field populations should be examined not only in the context of life-history trade-offs but, also, through the lens of abiotic stress resistance.

In the present study, we examine temperature and osmotic stress resistance in two strains originating from differential selection for insecticide resistance. The originating population, collected in the Yucatan Peninsula and insecticide-resistant, was reared in the lab for several generations without exposure to insecticide and then split into two strains, one selected for permethrin resistance and the other not. We took advantage of this selection regime to examine whether these sister strains originating from a common background differed significantly in several key abiotic stress resistance factors, as well as some life history traits. We hypothesized that selection for enhanced insecticide resistance should incur negative trade-offs in both abiotic stress resistance and life history traits. This is the first study to investigate the effects of selection for insecticide resistance on abiotic stress tolerance in *A. aegypti* and, in so doing, establishes straightforward protocols for such tests. Highlighting the possible links between abiotic stress resistance and insecticide resistance is of particular importance in the context of climate and land use changes, as possible interactive effects may help or hinder the spread of resistance genotypes to new locations [39].

## 2. Materials and Methods 

Eggs of two *A. aegypti* strains created from a population originally collected in the Vergel neighborhood of Merida, Yucatan, Mexico were obtained from the lab of W.C. Black IV at Colorado State University (Fort Collins, CO, USA). As a result of field selection by permethrin used in vector control programs in the region, the original population carried at least two mutations in the voltage-gated sodium channel that are known to confer knockdown resistance: a valine-to-isoleucine replacement in codon 1016 (V1016I) and a phenylalanine-to-cysteine replacement in codon 1534 (F1534C). The population was reared in the lab without exposure to insecticide for 8 generations, resulting in a loss of insecticide resistance. The F8 population was then split into two strains, Vergel-resistant (Vr) and Vergel-susceptible (Vs), with Vr exposed to 10–25 μg of permethrin every 3rd generation (to allow some recovery of fitness between bouts of selection) and Vs reared without selection. The eggs obtained for this experiment were from F29 (from the original date of field collection) for both strains. Bottle bioassay tests for permethrin LC50 on the F27, in which mosquitoes are placed in a glass bottle coated with insecticide for one hour and mortality is scored following a 24-h recovery time [26], found that Vr was 10-fold more resistant than Vs (Saavedra-Rodriguez, pers. com.). Random genetic drift should be minimal, as populations sizes were maintained at around 1000 individuals. We genotyped 108 female mosquitoes from the Vs strain and 108 from the Vr strain to assess the frequency of V1016I and F1534C mutations [40,41]. Briefly, we extracted DNA from individual mosquitoes following the methods described in [40] and performed two separate RT-PCRs (CFX96 Real-Time System, Bio-Rad, Hercules, CA, USA), each utilizing sets of primers corresponding to one or the other mutation. Expected haplotype frequencies of the combined V1016I and F1534C were calculated based upon the observed allele frequencies obtained from genotyping. For each of the 9 possible allelic combinations, the expected genotype frequencies were calculated considering the frequency of each allele from the haplotypes, so a mosquito that was VVFF (double homozygote-recessive) would have probability (pV × pV × pF × pF). These expected frequencies were then compared to the observed haplotype frequencies in order to test for linkage disequilibrium using a chi-square test.

For thermotolerance experiments, eggs were hatched, and larvae reared in clear deionized water in an incubator set at 26 °C and 12-h:12-h L:D (same conditions as they had been maintained in at Colorado State University). When larvae were 2 days old, they were split into replicate containers of 30 larvae / 350-mL water. The larvae were provided food daily (1:1 ground-up Tetramin^®^ fish flakes and Baker’s yeast). Thermotolerance of 4th instar larvae was examined by subjecting individuals to thermal stress for various durations and assessing survival 24 h later. For each trial, thirty 4th instar larvae from each strain were placed individually in 13 mm × 100 mm glass culture tubes with approximately 2 mL water. The 60 tubes were then immersed in a water bath set at 42 °C (Thermo Electron Corporation, Walmath, MA, USA). Preliminary tests found that larvae died within minutes at 44 °C but survived multiple hours at 40 °C. Therefore, 42 °C was chosen, as it promoted survival long enough to discriminate resistance to acute thermal stress while minimizing the possibility for additional variables (such as the production of heat shock proteins, for example) to affect the survival rates at the different time points. A thermocouple was placed in an extra culture tube to monitor the water temperature. Sets of 6 tubes per strain were removed from the water bath at 40, 50, 55, 60, and 70 min and then immediately placed in a room temperature water bath for 10 min to return the water temperature to ambient. Larvae were then pooled into new containers according to strain and stress duration, provided food, and left to recover overnight. The next day, larvae capable of diving and returning to the surface were counted as alive while those unable to do so were counted as dead. We performed 17 trials for this experiment.

For salinity tolerance experiments, eggs were placed (approximately 200 eggs per 350 mL container) in deionized water with either 0, 4, 8, or 12 g of aquarium salt per liter (corresponding to 0–37.5% sea water; Instant Ocean, Blacksburg, VA, USA). The upper salinity was chosen, because it is known to affect the survival of *A. aegypti* larvae without causing significant mortality [36]. Each day, new hatchlings were transferred to a separate container of the same salinity, and the hatching date was noted for each container. In total, we made 8 replicate larval containers of 19–22 larvae for each salinity and each strain (8 × 4 × 2 = 64 containers in total). Mosquitoes were monitored throughout larval development and then transferred to emergence cages on the day of pupation. Upon emergence, adult females were CO_2_-anesthetized, and wet mass was determined using a microbalance (Toledo XP26, Mettler, Columbus, OH, USA). Each mosquito was then dried at 70 °C for 24 h and weighed again to determine dry mass. We were careful to include representatives from all replicate containers and diverse larval durations in our adult mass measurements. Results are expressed as dry mass in mg and dry mass–specific (dms) body water content (by dividing water by dry mass) in order to standardize values for the variations in body mass among the mosquitoes. A total of 40 females were measured per salinity and strain.

Chi-square tests for linkage disequilibrium were performed by hand, while all other statistical tests were performed using the statistical software package SPSS version 25.0 (IBM SPSS Statistics, Armonk, NY, USA). Data were first tested for normality. For those data that did not fit a normal distribution (% larval survival relative to thermal stress and to salinity), Mann-Whitney U tests were used to examine the effects of strain on the dependent variable. In other cases, univariate general linear models were used with strain and salinity as the fixed factors. Bars in the figures represent the standard error of the mean (S.E.M.). In cases where significant interaction effects were detected between strain and salinity, we made pairwise comparisons between strains at each salinity using either *t*-tests or Mann-Whitney U tests, with a Bonferroni-corrected α of 0.0125. Salinities where a significant difference was found between strains are indicated with an asterisk in the figures.

## 3. Results

### 3.1. Strain Haplotypes

We found that the 1016I allele frequency was 0.16 in Vs and 0.85 in Vr, and the 1534C allele frequency was 0.17 in Vs and 0.85 in Vr. Both strains were found to be in Hardy-Weinberg equilibrium at both loci. Concerning the association between both loci, Vs was composed of 61% double-susceptible homozygotes (VVFF), while Vr was composed of 69% double-resistant homozygotes (IICC) (Figure 1). The next most common genotype, in both strains, was the double heterozygote (VIFC). The two loci were found to be in significant linkage disequilibrium in both strains (Table 1), with 1016V associated with 1534F and 1016I with 1534C.

### 3.2. Larval Thermotolerance

Overall, Vr larvae showed significantly higher survival than Vs larvae when exposed to a thermal stress of 42 °C, with median values of 68.7 and 37.5, respectively (total *N* = 170, U = 2101, Z = −4.776, *p* < 0.001; Figure 2). Assuming a linear change in survival over the timeframe of measurement, 50% mortality occurred at approximately 50 min for Vs and 64 min for Vr.

### 3.3. Larval Survival and Duration at Different Salinities

We found no overall significant difference in larval survival between both strains within the range of salinities tested, with median values of 95% for Vr and 100% for Vs (total *N* = 64, U = 551.5, Z = 0.570, *p* = 0.569; Figure 3a). The larval duration was not affected by the strain (degree of freedom (df) = 1, F = 0.159, *p* = 0.692); however, it was affected by the salinity (df = 3, F = 24.487, *p* < 0.001), and the effect was strain-dependent (df = 3, F = 4.437, *p* = 0.007; Figure 3b). Specifically, the larval development of Vs was delayed in both freshwater and 12 g·L^−1^, whereas the larval development of Vr was delayed only at the highest salinity. The delay in development at 12 g·L^−1^ was significantly stronger in Vr than in Vs (*p* = 0.010).

### 3.4. Adult Dry Mass and Water Stores of Females Reared at Different Salinities

Newly emerged females of both strains did not differ significantly in their overall dry mass (df = 1, F = 0.393, *p* = 0.531; Figure 4a). However, the dry mass was affected by salinity (df = 3, F = 22.272, *p* < 0.001), and the effect was strain-specific (df = 3, F = 14.901, *p* < 0.001). In particular, Vr had a significantly higher mass than Vs at extreme salinities, but Vs had a significantly higher mass than Vr at 4 g·L^−1^. Water reserves were significantly affected by both strain (df = 1, F = 22.069, *p* < 0.001; Figure 4b) and by salinity (df = 3, F = 7.911, *p* < 0.001), with the effect of salinity being strain-specific (df = 3, F = 5.084, *p* = 0.002). While the Vr water content was relatively unaffected by larval salinity, the susceptible strain showed a significantly higher body water content compared to Vr if larvae were reared either in freshwater or 12 g·L^−1^.

## 4. Discussion 

In the present study, we examined two mosquito strains originating from one location in Southern Mexico, then differentially selected for insecticide resistance. We measured the frequency of alleles at two loci known to associate with insecticide resistance and found them to differ significantly among the strains. Furthermore, we found the alleles at both loci to be in linkage disequilibrium. The vast majority of mosquitoes tested were either double homozygote-susceptible (for the Vs strain) or double homozygote-resistant (for Vr), and the next most common combination was double heterozygotes for both strains. Mutations at sites 1534 and 1016 of the VGSC cooccur widely throughout the Americas [42,43,44,45]. Hypotheses for the association include one compensating for the negative fitness effects of the other [24] or enhancing the insensitivity caused by the other [23]. The distribution pattern we found provides support for the concept of synergism between both. The strains studied here likely carry other resistance mutations that were not genotyped but may also influence the observed differences between strains.

The experiment presented here allowed us to compare fitness traits in two strains with identical genetic backgrounds but differentially selected for permethrin resistance over multiple generations. Contrary to our expectations, we found a significant positive effect of insecticide resistance selection on larval thermotolerance. *A. aegypti* is a container-dwelling species where larvae are restricted to small, shallow bodies of water such as abandoned tires [46], in which abiotic factors may vary widely for the duration of their development. Furthermore, the high specific heat and thermal conductivity of water ensure that the body temperature of the larva closely matches that of the surrounding water. Our findings indicate that insecticide resistance may correlate with enhanced larval survival under heat-stressed field conditions for *A. aegypti*. The mechanisms limiting insect survival at the upper thermal extremes are not fully understood but may involve multiple factors. For example, protein denaturation [47] may lead to a breakdown of neuromuscular or metabolic processes. Amino acid substitutions resulting from mutations either in genes directly involved in insecticide resistance or in those co-segregating with resistance genes may affect protein stability at high temperature [48]. It has also been hypothesized that, at very high temperature, oxygen delivery may become insufficient to meet metabolic demand [49]. Aquatic organisms that obtain oxygen directly from the water are indeed prone to oxygen deprivation at high ambient temperatures [50,51]. Mosquito larvae, however, obtain oxygen from the air via their siphon, so this factor is unlikely to dictate thermotolerance [52,53]. Other possible differences between strains include the management of reactive oxygen species produced during thermal stress [54,55], the homeoviscous adaptation potential of the cell membrane [56,57], or differential heat shock protein expression among the strains [58,59,60,61]. Heat Shock Protein STI1, for example, a protein that may act as a cochaperone mediating the activity of HSP70 and HSP90 [62,63], was found to genetically associate with insecticide resistance in *A. aegypti* [64]. Further studies will be necessary to help determine which physiological mechanisms set the upper thermal limits in mosquito larvae and how these are affected by selection for insecticide resistance.

To our knowledge, this was the first study to examine the impact of insecticide resistance selection on larval salinity tolerance in *A. aegypti*. While larval survival was not affected by selection for enhanced insecticide resistance, our study found a difference between strains in larval duration at extreme salinities. Specifically, compared to the susceptible strain, the resistant strain took longer to pupate at the highest salinity but less time at the lowest salinity. *A. aegypti* are freshwater osmoconformers who lack a salt-secreting rectal gland and are thus limited to environments more dilute than their hemolymph osmotic pressure [65]. Larval survival of *A. aegypti* is possible in fresh and brackish water [66] but drops rapidly beyond 10–15 g·L^−1^ [36,67]. Larval duration was shortest at 7 g·L^−1^, suggesting that the osmoregulatory cost may be lowest at a moderate salinity, allowing more energy to be dedicated to growth [36]. A short larval duration is a relevant fitness trait, as larvae may be at high risk of predation and evaporation of their habitat prior to pupation. The difference in larval duration between strains that we observed at extreme salinities may reflect an altered osmoregulatory ability, wherein the resistant strain shows a lower cost of osmoregulation in hypoosmotic conditions but a higher cost and lower fitness in the hyperosmotic medium. This result suggests that salinity tolerance may also be worth examining in the context of traits affecting the fitness of insecticide resistance genotypes.

As expected, we found that larvae grown at higher salinities produced adults of lower mass. Salinity was previously found to affect the growth rate and larval duration in a nonproportional manner, such that adult mass is negatively affected by a high larval salinity [36]. While no significant difference in dry mass was found between the strains overall, mass varied between strains within the salinity groups. In particular, the resistant strain had a higher dry mass in freshwater than the susceptible strain, despite showing a shorter larval duration. The combined rapid growth rate and large adult mass may represent again a selective advantage for the resistant strain in a typically inhospitable larval habitat. An increased body size has generally been thought of as a fitness advantage due to the fact that larger females can ingest more blood and produce more eggs [35]. However, a larger mass may also indicate a thicker cuticle, a feature that has direct benefits in conferring penetration resistance to insecticides [6,68,69] but may otherwise carry an increased cost during flight [70]. Future studies should include measurements of wing size in addition to body mass, as this metric would allow researchers to tease apart the relative contributions of body dimension and cuticular mass to the overall body mass.

The difference in mass-corrected body water content between strains at 0 g·L^−1^ might reflect a difference in relative cuticle mass. Indeed, if mosquitoes of both strains have similar tissue hydration, the resistant females may appear to have less water simply because a larger portion of their mass is composed of relatively dry cuticle. However, body water differences can also be explained by variation in energy storage strategies. The susceptible female may be storing most of her energy as glycogen, which has been shown to bind up to three times its weight in water [71]. Teasing apart the mechanistic reasons for the differences observed, and their potential fitness implications, will require measuring glycogen stores, as well as cuticular thickness, via transmission electron microscopy and/or expression patterns of cuticle-related cytochrome P450s and cuticular proteins involved in cuticle thickening. These investigations would also provide valuable information on the possible additional mechanisms of insecticide resistance in *A. aegypti*.

## 5. Conclusions

Selection for enhanced insecticide resistance in this population led to a series of unexpected changes, including at least one fitness benefit, increased larval thermotolerance, highlighting the association between insecticide resistance and abiotic stress resistance traits. Under field conditions, enhanced larval thermotolerance could result in a rapid selective sweep following a particular climate event driving the insecticide resistance of local populations closer to fixation. Larval habitat salinity may also selectively influence the fitness of alternate populations by affecting factors such as body size, energy storage, or cuticular thickness in the adults. Associations between insecticide resistance and other fitness traits may result from genetic linkages, possibly within chromosomal inversions [72] or micro-inversions [73], such as have already been observed in *Drosophila melanogaster* [74] and *Anopheles* spp. [75]. Understanding which fitness-related genes associate with insecticide resistance genes will not only provide crucial information on how weather and climate affect the insecticide resistance of field populations but, more importantly, the integration of that knowledge into management strategies could lead to more effective control of vector populations and ultimately reduce the spread of disease. 

## Figures and Tables

**Figure 1 insects-12-00124-f001:**
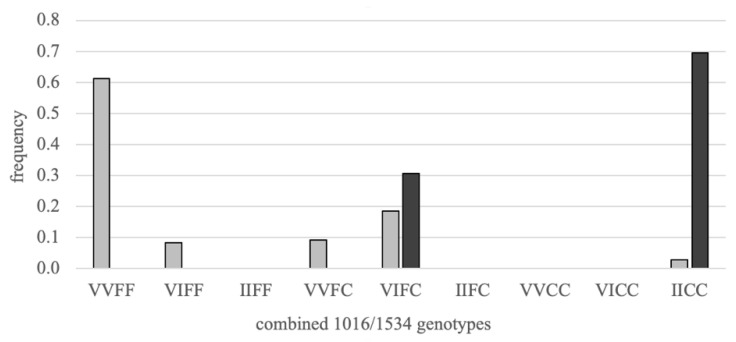
The frequency of 1016/1534 genotypes for both the susceptible and resistant strains used in the study. Randomly selected females from the susceptible (light bars) and resistant (dark bars) strains were genotyped for both V1016I and F1534C knockdown resistance (kdr) mutations. Double homozygote-susceptible individuals (VVFF) are on the far left and double homozygote-resistant (IICC) on the far right.

**Figure 2 insects-12-00124-f002:**
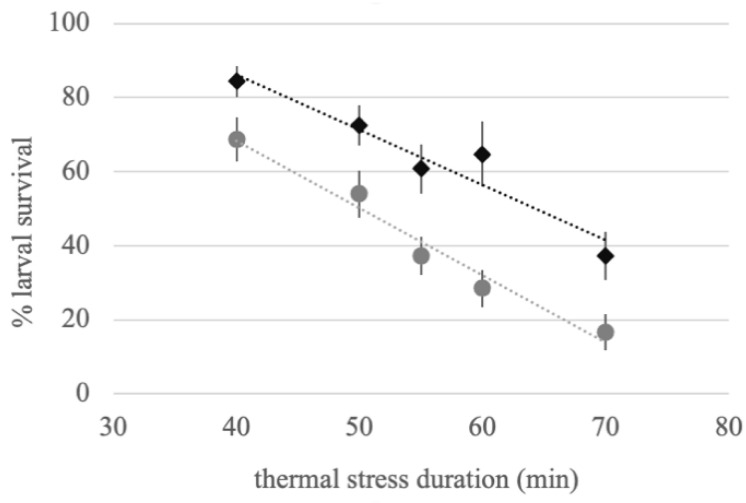
Effect of thermal stress on the larval survival of insecticide-resistant and -susceptible *Aedes aegypti* strains. Percentage of larvae still alive 24 h post-treatment following a 42 °C thermal stress of variable duration. The grey circles represent Vergel-susceptible (Vs), and the black diamonds represent Vergel-resistant (Vr). Bars represent S.E.M.

**Figure 3 insects-12-00124-f003:**
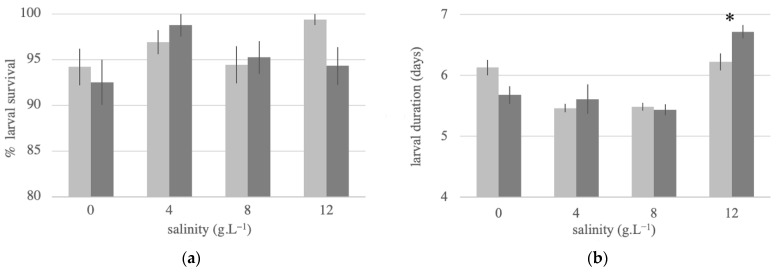
Effect of water salinity on the larval development success and duration. (**a**) Average percent of larvae that successfully pupated in each container. (**b**) Average number of days until pupation for larvae in each container. Light grey columns represent insecticide-susceptible larvae Vs, and dark grey columns represent insecticide-resistant larvae Vr. Bars represent S.E.M. Significant differences between strains within each salinity group are represented by an asterisk.

**Figure 4 insects-12-00124-f004:**
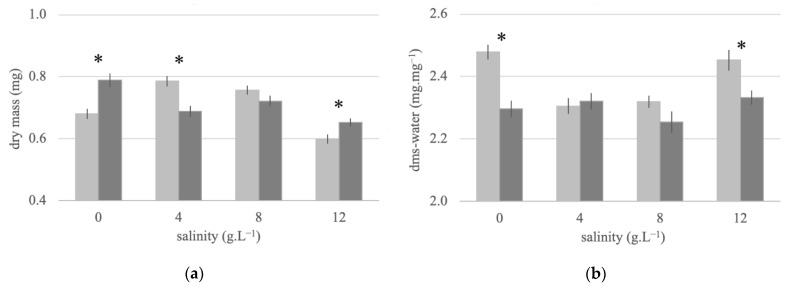
Body characteristics of newly emerged female mosquitoes from insecticide-susceptible (Vs) and -resistant (Vr) *A. aegypti* strains reared at different salinities. (**a**) Dry mass and (**b**) dry mass-specific water content. Light grey columns represent Vs adult females, and dark grey columns represent Vr adult females. Bars represent S.E.M. Significant differences between the strains within each salinity group are represented by an asterisk.

**Table 1 insects-12-00124-t001:** Observed and expected haplotype numbers in both strains, with the standardized disequilibrium value and test of significance of the linkage disequilibrium. Vs: Vergel-susceptible, Vr: Vergel-resistant. For haplotypes, the first letter represents the allele at the 1016 locus and the second letter represents the allele at the 1534 locus.

Strain	Haplotypes	Observed N	Expected N	D’	SumSquares	Chi-Square	*p*-Value
Vs	VF	322	302	0.35	1.37	50.75	<0.001
	VC	40	60		6.85		
	IF	38	58		7.09		
	IC	32	12		35.44		
Vr	VF	33	10	0.41	52.08	72.56	<0.001
	VC	33	56		9.39		
	IF	33	56		9.39		
	IC	333	310		1.69		

## Data Availability

Data available upon request.

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
