# Peer review of "Effect of Selection for Pyrethroid Resistance on Abiotic Stress Tolerance in Aedes aegypti from Merida, Yucatan, Mexico"

_insects, 2021, doi:10.3390/insects12020124_

Round 1

Reviewer 1 Report

This is a well-written paper that I enjoyed reading.  I have no major issues with the experimental design and feel that this is a tidy contribution to the integrated effects of insecticide resistance on vector physiology. I have a few minor comments below:

Line 149: Did you measure how quickly the vials returned to room temperature? “Quickly” is a bit vague. Unless you have a measurement of how long it took, I think just remove that qualifier.

Line 202: p = 569 – I’m assuming this is a typo!

Line 202-203: This sentence structure makes it sound a bit like larval duration and time to pupation are two different things. Maybe stick to one term?

Figure 3 – I think the y axis is slightly misleading – could you at least include an axis break? I think the effect size is a bit exaggerated.

For all figures, could you include letters to denote significance?

Lines 214-217: could you be a bit more specific here? Instead of just saying that things were affected/significant, could you provide the direction of change?

Line 289: I’m not suggesting you need to go back and do this, but perhaps mention (or consider for future!) that it will be useful to do both mass and wing size measures.

Line 290: reference for this last sentence?

Author Response

We thank Reviewer 1 for supportive and insightful comments that helped us improve the manuscript. We hope we have addressed the reviewer’s comments satisfactorily.

- Line 149: Did you measure how quickly the vials returned to room temperature? “Quickly” is a bit vague. Unless you have a measurement of how long it took, I think just remove that qualifier.

Response: we did monitor temperature change following removal of the vials from the water bath for several (but not all) replicates mostly to ensure that the rate of temperature change among vials was consistent, and that 10min was more than enough time to reach stable water temperature in the vials.  We agree that the term “quickly” is vague and unnecessary and therefore have removed it from the revised manuscript (line 161).

- Line 202: p = 569 – I’m assuming this is a typo!

Response: Yes it is a typo – thank you for catching that! We have fixed it.

- Line 202-203: This sentence structure makes it sound a bit like larval duration and time to pupation are two different things. Maybe stick to one term?

Response: we have changed “time to pupation” to “larval duration” here (line 226) and checked that “time to pupation” was not used elsewhere.

- Figure 3 – I think the y axis is slightly misleading – could you at least include an axis break? I think the effect size is a bit exaggerated.

Response: we replaced this and other figures to make axes less misleading.

- For all figures, could you include letters to denote significance?

Response: thank you for this suggestion. Because the contrast between strains was most important to our study, we have performed for each salinity either t-tests or UMW tests correcting for multiple measurements (Bonferroni-corrected alpha = 0.0125). For those salinities where the dependent variable was significantly different between strains, we added an asterisk.

- Lines 214-217: could you be a bit more specific here? Instead of just saying that things were affected/significant, could you provide the direction of change?

Response: upon suggestions of another reviewer, we have performed pairwise comparisons between strains for each salinity and added asterisks on the figures to denote significance. We have also expanded the descriptions within the results section to provide more detail on the direction of change.

- Line 289: I’m not suggesting you need to go back and do this, but perhaps mention (or consider for future!) that it will be useful to do both mass and wing size measures.

Response: absolutely, this is a very good point and a measure that, after the fact, we wish we’d done. We have added a note regarding this in lines 321-323.

- Line 290: reference for this last sentence?

Response: indeed – thank you, we have added a reference concerning the possible cost of increased flight resulting from added weight of the cuticle.

Reviewer 2 Report

Overall Notes

This is a well-done study and a good example of simple procedures and experimentation providing meaningful results. Moreover, as the authors outlined, this paper may provide a framework for reevaluating how we consider resistance mutations.

The authors sought to quantify the fitness effects for insecticide selection in Aedes aegypti against two abiotic stressors: temperature and salinity (osmotic balance). While previous experiments have found a trade-off between insecticide resistance and fecundity in mosquitoes, responses to abiotic factors have not been investigated. The authors hypothesized there would be negative fitness effects noted with insecticide resistance, however, the data showed increased thermotolerance and no significant difference in pupation success at varying salinities.

The general conclusions in this study are supported by the data. We have some recommendations that should be addressed before this manuscript is ready for publication.

Major comments.

Introduction

The introduction is well written and gives a complete background to believe the authors should provide justification for the choice of abiotic factors over many others, as well as explain the temperature and salinity stress values. Are these values realistic or expected to occur under natural conditions? If the vgsc is a single copy gene and to mention its accession number.

Methods

Although the authors mentioned this study provides “repeatable protocols” (Line 110), there is a lack of details on how many experiments were done.  More details on procedures and rationale will increase the repeatability.

The authors state that the original population contained kdr mutations. Were these developed due to spraying of permethrin, the same insecticide used in the selection process ( (L124) or another insecticide?

The authors report that selection with insecticides was done every third genetraton (L124). Is there a reason for this?  

Was the difference (10 fold) in resistance between Vs and Vr an expected result? In the field would this be expected, is this low, or high? Should these data be provided in the results section?

What were the conditions under which mosquitoes were raised in the lab before experimentation. If these were the same conditions indicated on line 139, please indicate this.   

The thermal tolerance experiments measured the effects of moving larvae from the comfort of 26C directly into 42C. This is more likely thermal shock than tolerance. The authors should address this in this section and in the discussion. Under normal conditions the ambient temperatures for the larvae will increase and decrease gradually, or quickly, but not normally a change of 16C in 30 seconds.      

Please explain in more detail the genotyping procedures and the data and methods used to test the levels of resistance.  The bottle assays are standard procedures  to test insecticide resistance, but these data are essential to this study and should be included or cited if used in any other study.

It is not clear how the statistical analyses were done. Looking at the results, it is not evident if the reported p-values correspond to GLM analyses, U-tests, or any other tests.

Results.

The data are presented with inconsistent formats. Survival to thermal stress in Figure 1 is presented as continuous data as a line graph. There should be more consistency in data presentation. Was this done because the data in Figure 1 was assumed to be linear? Why was this not assumed for the remaining data?

It is confusing to see in Figure 2 to be and 3 a continuous variable (salinity) represented in columns as if it was a discrete variable. This representation suggests pairwise analyses were done instead of a GLM global analysis. In the methods, it is mentioned that Figure 4a was analyzed using a GLM (Lines 170-171), but the results on Lines 213-214 suggest that a pairwise test was done.

The authors could test all the data with GLM models first, then perform paired T-test or UMW tests corrected for multiple measurements, and report both analyses.

Please show a graph with the data for time to pupation (Line 202). This is mentioned to be measured but not shown.  

Specific Changes

Line 70: add the accession id for this gene

Line 199: S.E.M was not outlined in the materials and methods as a statistical definition.

Figure 3: From my understanding, the authors are comparing within the groups, yet the bars are paired together as if they are comparing between the groups. If my interpretation is correct, then I suggest putting all Vs bars together and all Vr bars together. Same for Figure 3b.

Author Response

We thank Reviewer 2 for detailed feedback and for raising questions that helped us improve the manuscript. We hope we have addressed the reviewer’s comments and questions satisfactorily.

Major comments.

Introduction

  • The introduction is well written and gives a complete background to believe the authors should provide justification for the choice of abiotic factors over many others, as well as explain the temperature and salinity stress values. Are these values realistic or expected to occur under natural conditions? If the vgsc is a single copy gene and to mention its accession number.

 Response: Thank you for these comments. Such high temperatures may well occur, however we have not measured habitat temperatures in the field. However, our purpose here was not to mimic natural conditions but rather utilize a fairly standard thermal tolerance assay to compare survival of alternative strains. Our choice of 42°C specifically was the result of preliminary experiments which demonstrated that this resulted in about 50% mortality in a timeframe that was slow enough to discriminate between strains (at 44°C larvae died within minutes) but rapid enough to minimize effects of other factors (at 40°C larvae survived multiple hours, a sufficient amount of time for synthesis of heat shock proteins for example). (see lines 154-158 for additional justification).

Ae. aegypti are known from previous studies to survive in water up to about 15 g/L salinity, however we haven’t tested the salinity of water in the environment where this particular population originated. We decided to compare survival over a range we knew the larvae would survive but experience a variable amount of stress. We have provided additional justification of salinity values we used in the methods section (lines 168-170).

We have added the accession number of the vgsc gene (line 75-76).

Methods

  • Although the authors mentioned this study provides “repeatable protocols” (Line 110), there is a lack of details on how many experiments were done.  More details on procedures and rationale will increase the repeatability.

Response: Thank you for this point. By “repeatable protocols” we meant to suggest that the experiments performed here were straightforward enough that they could be used again for comparing other populations. We have rephrased this, by replacing “repeatable” with “straightforward” (line 111). We also provide additional descriptions of the experiments done in the methods section (see below for details).

  • The authors state that the original population contained kdr mutations. Were these developed due to spraying of permethrin, the same insecticide used in the selection process (L124) or another insecticide?

Response:  Permethrin had been used during a long-term vector control campaign in the region surrounding Merida, Mexico at the time the population was collected. We have added this detail in lines 119-120.

  • The authors report that selection with insecticides was done every third generation (L124). Is there a reason for this?  

Response: indeed, the reason is that the selection process causes a large decrease in overall fitness, so the population is allowed a couple generations to recover between bouts of selection. We have provided this clarification in line 127.

  • Was the difference (10 fold) in resistance between Vs and Vr an expected result? In the field would this be expected, is this low, or high? Should these data be provided in the results section?

Response: The 10-fold change in resistance following 11 generations of differential selection was not a surprising result – a more recent study found some field populations to be 60 times more resistant than strains that haven’t been exposed since 2005 (Vera-Maloof et al 2020). We chose purposefully not to go into more detail on the nature and effects of the selection protocol as this was not the purpose of this particular study. Rather, we aimed to focus on our question regarding physiological responses to abiotic stress in 2 strains strongly differing in resistance genotype frequency.

  • What were the conditions under which mosquitoes were raised in the lab before experimentation. If these were the same conditions indicated on line 139, please indicate this.

Response: yes indeed we maintained them in the same conditions as they had been previously at CSU. We now specify that in lines 146-147.

  • The thermal tolerance experiments measured the effects of moving larvae from the comfort of 26C directly into 42C. This is more likely thermal shock than tolerance. The authors should address this in this section and in the discussion. Under normal conditions the ambient temperatures for the larvae will increase and decrease gradually, or quickly, but not normally a change of 16C in 30 seconds.

Response: you are correct, it is true that under natural conditions such a rapid temperature change is not likely. However, our purpose here was not to mimic natural conditions but rather utilize a fairly standard thermal tolerance assay to compare survival of alternative strains. Our choice of 42°C specifically was the result of preliminary experiments which demonstrated that this resulted in about 50% mortality in a timeframe that was slow enough to discriminate between strains (at 44°C larvae died within minutes) but rapid enough to minimize effects of other factors (at 40°C larvae survived multiple hours, a sufficient amount of time for synthesis of heat shock proteins for example). (see lines 154-158 for additional justification).

  • Please explain in more detail the genotyping procedures and the data and methods used to test the levels of resistance.  The bottle assays are standard procedures to test insecticide resistance, but these data are essential to this study and should be included or cited if used in any other study.

Response: We have provided additional referenced detail on the genotyping procedures in lines 135-138. Also, we have provided a referenced description of the bottle bioassay in lines 129-132.

  • It is not clear how the statistical analyses were done. Looking at the results, it is not evident if the reported p-values correspond to GLM analyses, U-tests, or any other tests.

Response: we have added more statistical detail in the results section, including degrees of freedom, sample size, U- or F-value to clarify the nature of tests done (see lines 214-215; 225-228; 237-243).

Results.

  • The data are presented with inconsistent formats. Survival to thermal stress in Figure 1 is presented as continuous data as a line graph. There should be more consistency in data presentation. Was this done because the data in Figure 1 was assumed to be linear? Why was this not assumed for the remaining data?

Response: Survival to thermal stress was assumed to be directional, with longer duration leading to higher mortality, therefore we felt comfortable presenting these data as a line graph. On the other hand, we did not expect salinity to affect dependent variables in such a directional manner (A aegypti actually do better at low salinities compared to complete freshwater), and we felt uncomfortable extrapolating between our four salinity groups. This is the reason we chose to present all salinity data as columns.

  • It is confusing to see in Figure 2 to be and 3 a continuous variable (salinity) represented in columns as if it was a discrete variable. This representation suggests pairwise analyses were done instead of a GLM global analysis. In the methods, it is mentioned that Figure 4a was analyzed using a GLM (Lines 170-171), but the results on Lines 213-214 suggest that a pairwise test was done.

Response: We hope our explanation above helps to clarify why we chose to present the salinity data as discrete rather than continuous. For statistical analyses, all salinity data that were confirmed to be normally distributed were tested using a GLM with salinity and strain as fixed factors. We have added statistical values relevant to the tests used for all reported p-values which should clarify which tests were used where.

  • The authors could test all the data with GLM models first, then perform paired T-test or UMW tests corrected for multiple measurements, and report both analyses.

Response: thank you for this suggestion. We have performed either t-tests or UMW tests correcting for multiple measurements (Bonferroni-corrected alpha = 0.0125). For those salinities where the dependent variable was significantly different between strains, we added an asterisk.

  • Please show a graph with the data for time to pupation (Line 202). This is mentioned to be measured but not shown.  

Response: we mistakenly used the term “time to pupation” instead of “larval duration” (line 226). We have changed the terminology in the text to match Figure 3b which presents these data.

Specific Changes

  • Line 70: add the accession id for this gene

Response: thank you, this information was added in line 74.

  • Line 199: S.E.M was not outlined in the materials and methods as a statistical definition.

Response: thank you for pointing out this omission. We have added this on lines 188-189.

  • Figure 3: From my understanding, the authors are comparing within the groups, yet the bars are paired together as if they are comparing between the groups. If my interpretation is correct, then I suggest putting all Vs bars together and all Vr bars together. Same for Figure 3b.

Response: thank you for raising this point. We were unsure of how to present these data. While the within strain comparison is interesting, our main goal was to examine whether strains differed in the physiological parameters tested. We felt that pairing strains while using contrasting greys would allow the reader to compare both within and between relatively well. Following this reviewer’s suggestion, we performed pairwise comparisons and indicated significant differences between strains on the figures using asterisks.  

Reviewer 3 Report

The authors studied the “Effect of selection for pyrethroid resistance on abiotic stress tolerance in Aedes aegypti from Merida, Yucatan, Mexico”. The manuscript is well-organized, and I am satisfied with the experimental part. However, the authors did not explain statistical methods in each result obtained in the experiments. The manuscript is in need of revisions before it is acceptable for publication. Please see my specific comments below:
L.36: Keywords should be in alphabetic order. Also, keywords serve to widen the opportunity to be retrieved from a database. To put words that already are into title and abstracts makes KW not useful. Please choose terms that are neither in the title nor in abstract.
L.46:...and behavioral adjustments…Please, explain.
Ls.63,64,70: For these insect species, provide ID identifier, order and family taxa.
Ls.167-171: What was the number of replicates for each experiment?
Ls.174-180 and Figure 1: Results do not show a statistical analysis. Please provide statistical details in material and methods section.
Table 1: How were the chi-square values? It is assumed that the data were analyzed with the Mann-Whitney U-test, presuming t-values in each experiment.
Ls.190-199: The data were analyzed using a linear regression? If so, more details about statistical method are needed in material and methods section.
Ls.200-206 and 211-217: Please, provide the t-value and degree of freedom for each statistical data (p< or p=) obtained.
Ls.312-316: This information should be in discussion session.

Author Response

We thank Reviewer 3 for pointing out specific issues with our manuscript and helping us improve it. We hope we have addressed the reviewer’s comments satisfactorily.

  • 36: Keywords should be in alphabetic order. Also, keywords serve to widen the opportunity to be retrieved from a database. To put words that already are into title and abstracts makes KW not useful. Please choose terms that are neither in the title nor in abstract.

Response: We have modified the keywords accordingly.

  • 46:...and behavioral adjustments…Please, explain.

Response: thank you for pointing this out. Upon further examination of the literature, we decided to remove the term “behavioral adjustments” as no evidence actually exists to support the hypothesis that behavioral avoidance of insecticide can be selected for. We have also removed the reference (line 43).

  • 63,64,70: For these insect species, provide ID identifier, order and family taxa.

Response: we have added order and family taxa for these species (lines 63, 65, 71).

  • 167-171: What was the number of replicates for each experiment?

Response: we made 8 containers for each salinity and each strain (lines 171-173). We did not have replicate measures for adult mosquitoes as the females were simply anesthetized and used upon emergence from whichever container they emerged from. We were careful to include mosquitoes from different containers and larval durations for our mass measurements so as not to bias our results. We have added this detail in lines 177-179.

  • 174-180 and Figure 1: Results do not show a statistical analysis. Please provide statistical details in material and methods section.

Response: no statistics were done for the data in Figure 1 as the figure reports the number of mosquitoes we identified with different genotypes in each strain. For other data, we have expanded the descriptions of statistical analyses both in the methods and results sections according to this and other reviewer’s comments (lines 183-193; 214-215; 225-228; 237-243).

  • Table 1: How were the chi-square values? It is assumed that the data were analyzed with the Mann-Whitney U-test, presuming t-values in each experiment.

Response: we indicate on lines 143-144 that expected and observed frequencies were compared using a Chi-square test. In order to improve clarity, we have provided additional detail within the last paragraph of the methods section (lines 183-184) where other statistical tests are described.

  • 190-199: The data were analyzed using a linear regression? If so, more details about statistical method are needed in material and methods section.

Response: For statistical analyses, all salinity data that were confirmed to be normally distributed were tested using a GLM with salinity and strain as fixed factors. We have added statistical values relevant to the tests used for all reported p-values which should clarify which tests were used where, and added more detail concerning statistical methods used in the methods section (lines 183-193).

  • 200-206 and 211-217: Please, provide the t-value and degree of freedom for each statistical data (p< or p=) obtained.

Response: thank you for pointing out this omission – we have added statistical values relevant to the tests used for all reported p-values.

  • 312-316: This information should be in discussion session.

Response: thank you for pointing this out; we have rephrased this information to hopefully fit better within the conclusion section. We thought it more appropriate to keep the mention of co-segregation of alleles outside of the actual discussion, as this is not within the scope of our study.